# Typhoon-Induced Fragility Analysis of Transmission Tower in Ningbo Area Considering the Effect of Long-Term Corrosion

Qiang Li [1,2,*], Hongtao Jia [3], Qing Qiu [4], Yongzhu Lu [5], Jun Zhang [1], Jianghong Mao [6], Weijie Fan [1] and Mingfeng Huang [7]

1   School of Civil Engineering and Architecture, NingboTech University, Ningbo 315100, China; zj@nit.zju.edu.cn (J.Z.); fanwj@nit.zju.edu.cn (W.F.)
2   Zhejiang Engineering Research Center for Intelligent Marine Ranch Equipment, Ningbo 315100, China
3   School of Civil Engineering, Chongqing Jiaotong University, Chongqing 400074, China; jiahongtao@nblg.wecom.work
4   Ningbo Kaihong Engineering Consulting Co., Ningbo 315500, China; qq854410776@hotmail.com
5   Bank of Ningbo Co., Ningbo 315100, China; luyongzhu@nbcb.cn
6   College of Architecture & Environment, Sichuan University, Chengdu 610021, China; jhmao@scu.edu.cn
7   Institute of Structural Engineering, Zhejiang University, Hangzhou 310058, China; mfhuang@zju.edu.cn
*   Correspondence: liqiang@nit.zju.edu.cn; Tel.: +86-1373-2253-784

**Featured Application: There are more and more large-span transmission projects with high towers and complex structures along the southeast coast of China, which are inconvenient for detecting corrosion during the service life and cause difficulties in repair and replacement. The work of this paper can provide an important reference for local power maintenance departments to carry out wind-resistant assessments of corroded transmission towers.**

**Abstract:** The purpose of this paper was to investigate the influence of long-term corrosion on the deterioration of wind resistance of a steel transmission tower during its service life. An analytical model for predicting the long-term corrosion depth of carbon steel was established, and the corrosion depth of carbon steel in the Ningbo area was predicted based on the local atmospheric environment data. With the help of typhoon full-track simulation and wind field simulation technology, a joint probability distribution model of multidirectional extreme wind speeds was constructed using the t-Copula function to determine the typhoon climate of the transmission tower site. Finite element models of the ZM4 cathead transmission tower under 30/60/90 corrosion years were then established, respectively, according to the predicted corrosion depth of carbon steel in Ningbo. Three damage modes, i.e., minor damage, moderate damage and severe damage, corresponding to the transmission tower under wind loads, were defined, and pushover analyses were used to determine the limit values of each damage mode so as to obtain the typhoon-induced fragility curves of the transmission tower within 30/60/90 corrosion years. The results show that the increase in corrosion age leads to a deterioration in the nominal mechanical properties of the transmission tower components, making the damage probability to the transmission tower increase. Under the typhoon wind loads of a 50-year return period in the most unfavorable wind direction in Ningbo, the probability of moderate damage of the tower is within 10% and the probability of minor damage is controlled between 10% and 40%.

**Keywords:** typhoon; transmission tower; corrosion; fragility analysis; extreme wind speed

## 1. Introduction

Ningbo is an important industrial port city in the southeast of China, with dual environmental characteristics of strong corrosion and strong typhoons. Its industrial–marine atmospheric environment will accelerate the corrosion of the galvanized layer on the surface of local steel structures, which, in turn, can lead to steel corrosion. According to the

current national standard, "Corrosivity Classification of Atmospheric Environment" [1], the atmospheric environment is mainly divided into an urban atmosphere, rural atmosphere, marine atmosphere and industrial atmosphere. Among them, the marine atmosphere and industrial atmosphere are typical corrosive atmospheric environments. In the marine atmosphere (the corrosion mechanism is shown in Figure 1), the high concentration of chloride ions is the main factor causing the corrosion of carbon steel, dissolved in the liquid film of chloride ions and greatly enhancing the conductivity of the liquid film solution, thus intensifying the polarization reaction and making the corrosion rate increase [2]. In the industrial atmosphere (the corrosion mechanism is shown in Figure 2), sulfur dioxide and other corrosive gases are the main factors causing the corrosion of carbon steel; these acidic gases dissolved in the liquid film makes the pH value of the solution lower, thereby increasing the corrosion of carbon steel [3]. There are more and more large-span transmission projects with high towers and complex structures along China's eastern coast, which are inconvenient for detecting corrosion during the service life and cause difficulties in repair and replacement. Steel corrosion not only reduces the cross-sectional area of steel members but also decreases the remaining mechanical properties of steel. Therefore, the accurate prediction of the carbon steel corrosion depth becomes an important prerequisite for the performance assessment of corroded transmission towers in service.

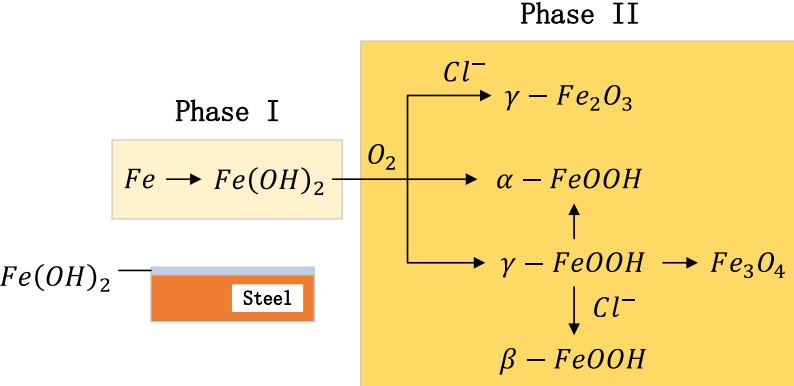

**Figure 1.** Marine atmospheric corrosion mechanism of steel.

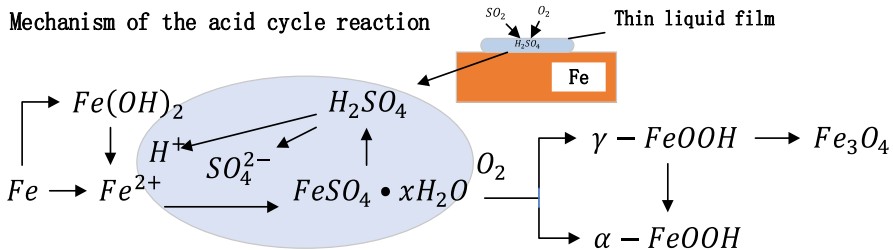

**Figure 2.** Industrial atmospheric corrosion mechanism of steel.

Starting from the 1950s, countries such as the United States, the United Kingdom, the Soviet Union and Japan have successively started tests on the natural exposure of metals and nonmetals to the atmosphere. Up to now, more than 400 atmospheric corrosion test stations have been established all over the world. Based on a large number of atmospheric exposure data, various types of atmospheric corrosion maps, relevant standards for atmospheric exposure tests and evaluation methods, have been introduced [4,5], and these results have greatly promoted the development of anticorrosion materials and protection technologies and enabled researchers to have a new understanding of atmospheric corrosion mechanisms and laws.

At present, in terms of the fragility analysis of the transmission tower, structural fragility under seismic loading has been widely studied [6–8], but research on wind-induced fragility is still relatively rare. In particular, the wind-induced fragility analysis of

the transmission tower, considering the effects of the long-term corrosion of steel members, is almost nonexistent. Xiao [9] analyzed the wind-induced fragility of the transmission tower based on a pushover analysis and incremental dynamic analysis (IDA) and studied the collapse resistance of the transmission tower. Ge [10] generated the wind load series of the transmission tower based on the linear filter method and conducted a wind-induced fragility study of the transmission tower to determine the wind resistance performance. Fu et al. [11] studied the structural fragility of the transmission tower under the joint action of wind and rain loads. Huang et al. [12] proposed a Bayesian approach to assess the discharge failure probability of overhead transmission lines under typhoon hazards. Based on full-scale measurement data collected during the two strong typhoons of Talim (1718) and Kong-rey (1825), the wind-induced fragility of a full-scale transmission line was updated using the proposed approach.

In addition, Ningbo is seriously affected by typhoon disasters every year. For example, the Super Typhoon Lekima hit Ningbo in 2019, causing 408 houses to collapse and 1594 houses to be damaged citywide. It is necessary to carry out a typhoon-induced fragility analysis for local corroded transmission towers to ensure safety during operations. Benefiting from the rapid growth of the Monte Carlo method and computer technology, extreme typhoon wind speed simulations have mushroomed in the past twenty years, which is mainly composed of two parts: (1) typhoon track simulation and (2) wind field simulation. Vickery et al. [13] pioneered a full-track method to generate synthetic hurricane tracks from genesis to lysis based on the historical track records in the National Hurricane Center's North Atlantic hurricane database (HURDAT). The development and utilization of the full-track models for wind hazard assessments have since been considered and expanded (Powell et al. [14]; James and Mason [15]; Emanuel et al. [16]; Lee and Rosowsky [17]; Vickery et al. [18]; Li and Hong [19,20]; Chen and Duan [21]). So far, storm tracks can be synthesized rapidly from purely statistical intensity algorithms. However, the effects of natural or anthropogenic climate changes could not be encompassed through the above empirical models. Today, some novel intensity models considering environment variables that can be obtained from reanalysis or global climate models have the potential to estimate future wind hazards under future climate projections. Jing and Lin [22] developed a hidden Markov model (MeHiM), which is dependent on the surrounding large-scale environment, such as vertical wind shear, relative humidity and ocean feedback from the reanalysis to simulate the whole process of hurricane intensity evolution. Huang et al. [23] verified the applicability of MeHiM in the Northwest Pacific Ocean and presented a general framework of a typhoon full-track simulation. For engineering applications, the wind field model can be classified as the gradient wind field model [17] and the planetary boundary layer (PBL) model [18,19,24]. Meng et al. [24] proposed an analytical model with an upper inviscid layer of cyclostrophic balance and a lower friction layer to calculate the wind field in a moving typhoon boundary layer. Thompson and Cardone [25] upgraded a PBL model by increasing the spatial resolution to simulate a wider variety of radial pressure and wind profile forms. These PBL models have been widely applied in the assessment of hurricane/typhoon wind hazards for the coastal regions of the United States and China [19,20].

This paper focuses on the analysis of the influence of long-term corrosion on the deterioration of wind resistance of a steel transmission tower during its service life under the atmospheric environment in the Ningbo area. The corrosion depth of carbon steel in Ningbo area was predicted based on the local atmospheric environment data. With the help of a typhoon full-track simulation and wind field simulation technology, a joint probability distribution model of multidirectional extreme wind speed was constructed using the t-Copula function to determine the typhoon climate of a transmission tower site. Finite element models of the ZM4 cathead transmission tower under various corrosion years were then established according to the predicted corrosion depth of carbon steel in Ningbo. Three damage modes, i.e., minor damage, moderate damage and severe damage, corresponding to the transmission tower under wind loads, were defined, and a pushover

analysis was used to determine the limit values of each damage mode so as to obtain the typhoon-induced fragility curves of the transmission tower under various corrosion years.

## 2. Corrosion Depth Prediction of Carbon Steel

### 2.1. Construction of Predictive Model

In this paper, carbon steel corrosion data from different atmospheric exposure tests worldwide were used as samples. The data sets were obtained from the ISO-CORRAG program [26] and China Gateway to Corrosion and Protection (http://www.ecorr.org/, accessed on 1 July 2021), including the latitude and longitude of the exposure site, the corrosion depth of the carbon steel and the corresponding natural exposure time. According to the different geographical locations of the exposure sites (as shown in Figure 3), the atmospheric environments were divided into a marine atmosphere (M), industrial atmosphere (I), urban atmosphere (U), rural atmosphere (R), urban–marine atmosphere (U–M) and industrial–marine atmosphere (I–M). Exposure tests to the carbon steel atmosphere at some of the sites are shown in Figure 4. The exposure test protocols were designed in strict compliance with "ISO 9226:2012, Corrosion of Metals and Alloys" [26].

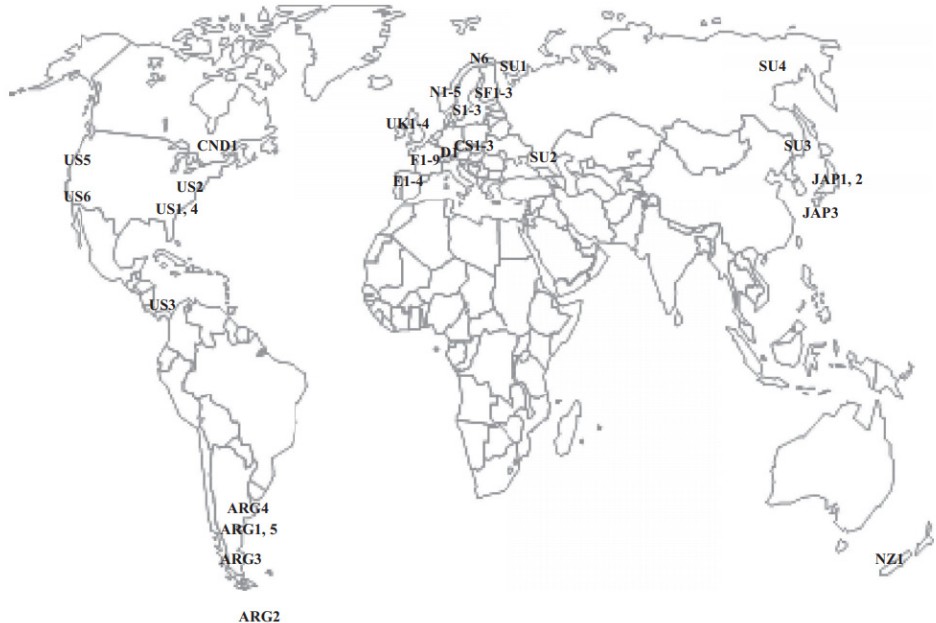

**Figure 3.** Geographic distribution of atmospheric exposure sites in the ISO-CORRAG program.

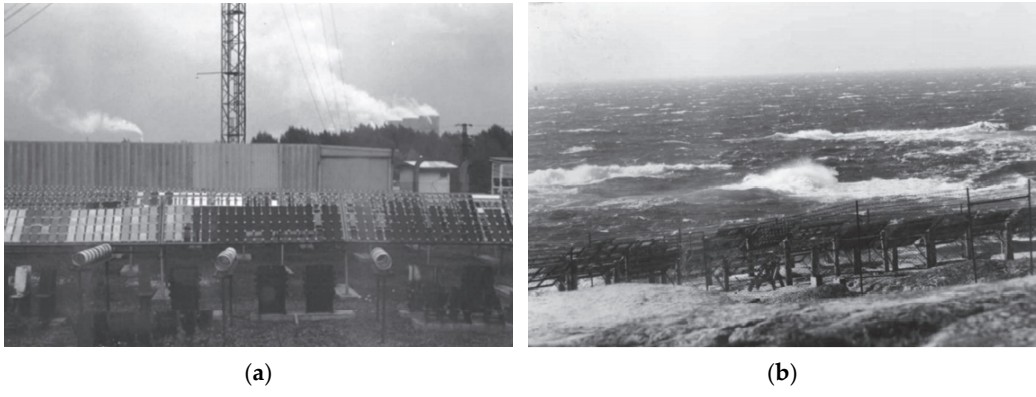

| (a) | (b) |
|---|---|

**Figure 4.** Atmospheric exposure test. (**a**) Industrial atmosphere. (**b**) Marine atmosphere.

The corrosion depth *D* (μm) was used to represent the corrosion loss of steel. Scholars have proposed various time-varying models for corrosion, mainly including linear, power,

logarithmic, exponential, nonlinear–linear and bimodal models [3]. However, according to the results of the long-term exposure tests, it was found that the corrosion loss of metals under different atmospheric environments basically follows the power–linear function law [27]: In the first stage, the metal corrosion loss shows a power function growth law; in the second stage, the corrosion loss grows linearly, i.e., it shows a stable development trend. The expressions are as follows:

$$\begin{cases} D_1 = D_0 \times t_1{}^n \\ D_2 = D_1 + \alpha t_2 \end{cases} \tag{1}$$

where $D_1$ indicates the first stage corrosion depth (μm), $D_2$ indicates the second-stage corrosion depth (μm), $D_0$ indicates the first-year corrosion depth (μm) and $t_1$ and $t_2$ indicate the first-stage and second-stage corrosion ages (year), respectively. According to the existing research [8] conclusions, carbon steel enters into the corrosion stabilization stage (second stage) when its corrosion age is 6 years, $\alpha$ indicates the second-stage corrosion rate (μm/year), $\alpha = d(D_2 - D_1)/dt_2$ and $n$ is a coefficient that characterizes the protective properties of corrosion products. According to the Tammann theory [27], the metal corrosion rate is controlled by the rate of oxygen across the corrosion products (rust layer) to the metal surface, so the thicker the rust layer, the more difficult it is for oxygen to reach the metal surface, and the smaller the corrosion rate.

Considering the correlation between $n$ values on the corrosiveness of the atmospheric environment, double logarithmic fits were performed according to the test data of each exposure site to obtain the $n$ values and the corresponding $D_0$ values. Figure 5 shows a double logarithmic plot of the corrosion depth vs. exposure time for carbon steel at some typical sites—specifically, marine (M) atmosphere for Tokyo, rural (R) atmosphere for Ahtari, urban (U) and industrial (I) atmospheres for Otaniemi and urban–marine (U–M) and industrial–marine (I–M) atmospheres for Ponteau Martigues. It can be seen that the corrosion rate of carbon steel is the fastest, and the first-year corrosion depth $D_0$ is also the largest for the exposure site in the urban–marine and industrial marine atmospheres. The first-year corrosion depth of carbon steel at the exposure site in the rural atmosphere is the smallest. The corresponding $n$ and $D_0$ values under each exposure site are shown in Figure 6. According to the specification GB/T19292.1-2018 [28] on the division of corrosion grade regions, the $D_0$ values under each exposure site in Figure 6 can be classified to the C1, C2, C3, C4 and C5 regions. It can be seen that the corrosion grade of carbon steel under the rural atmosphere was mainly concentrated in the C2 region, while the corrosion grades of carbon steel under the marine, urban/industrial and industrial–marine/urban-marine atmospheric environments were mainly concentrated in the C3 and C4 regions. In particular, the corrosion grades of carbon steel under the marine atmospheric environment, industrial–marine/urban–marine atmospheric environment at individual sites reached the C5 region. Overall, the $n$ value and the $D_0$ value shows a power function change law—that is, as the $D_0$ value increases, the $n$ value shows an obvious downward trend, the specific fitting function as shown in Equation (2), which is quite different from the specification of GB/T 24513.2-2010 [29], which provides a unified $n$ value under all atmospheric environments, indicating that this method of determining the $n$ value is more accurate.

$$\begin{cases} n_M = 2.1152 \times D_0^{-0.361} \\ n_R = 1.0903 \times D_0^{-0.287} \\ n_{I,U} = 0.8356 \times D_0^{-0.177} \\ n_{I-M,U-M} = 0.7145 \times D_0^{-0.087} \end{cases} \tag{2}$$

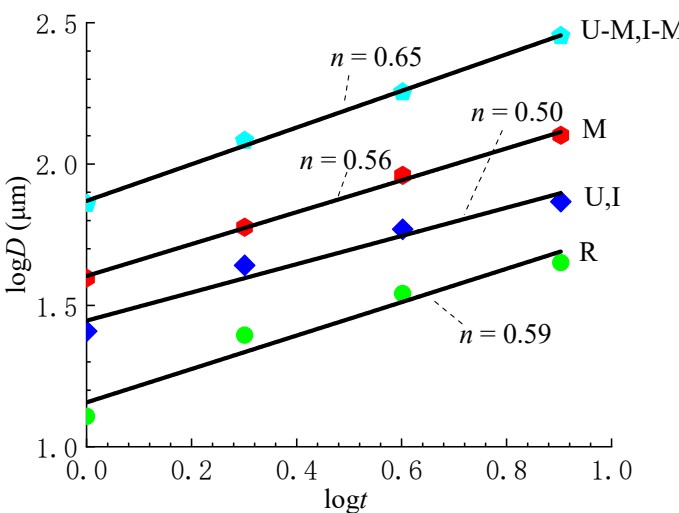

**Figure 5.** Double logarithmic plot of the corrosion depth vs. exposure time for carbon steel at some typical sites.

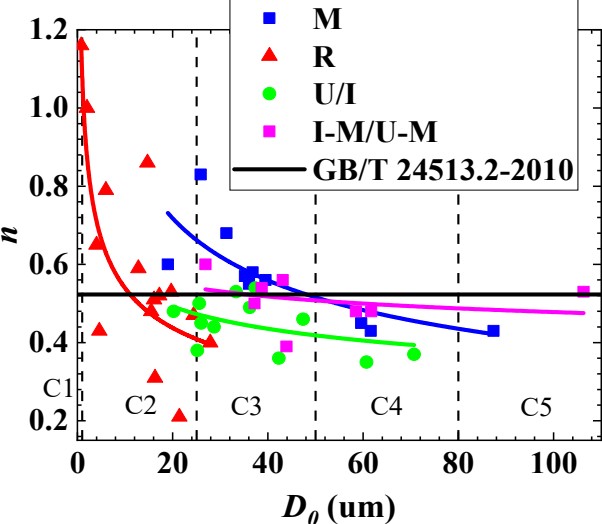

**Figure 6.** The corresponding $n$ and $D_0$ values under each exposure site.

### 2.2. Prediction of Corrosion Depth of Carbon Steel in Ningbo Area

Figure 7 shows the division of the atmospheric environment in each district and county of Ningbo. The Zhenhai District and Beilun District were classified as industrial–marine (I–M) atmospheric environments due to the heavy concentration of industry and the existence of large petroleum refineries. Cixi City and Xiangshan County are close to the sea and were classified as marine (M) atmospheric environments. Ningbo's main urban area, Yinzhou District, Haishu District and Jiangbei District, were classified as urban/industrial (U/I) atmospheric environments. Yuyao City, Fenghua District and Ninghai County were classified as rural (R) atmospheric environments due to their proximity to the mountainous area and a weak industrial base.

According to the specification in GB/T 24513.1-2009 [30], the first-year corrosion depth $D_0$ of carbon steel can be estimated according to the following formula:

$$D_0 = 1.77 P_d^{0.52} \times \exp(0.02RH + f_{St}) + 0.102 \times S_d^{0.62} \times \exp(0.033RH + 0.04T) \quad (3)$$

where $T$ denotes the annual average temperature (°C), $RH$ denotes the annual average relative humidity (%), $P_d$ denotes the annual average $SO_2$ deposition rate (mg/(cm$^2 \cdot$ d)), $P_d = 0.8 P_c$, $P_c$ is the $SO_2$ concentration (μg/cm$^3$), $S_d$ denotes the annual average $Cl^-$

deposition rate (mg/(cm$^2 \cdot$ d)) and $f_{St}$ denotes the carbon steel correlation coefficient; when $T \leq 10\,^\circ$C, $f_{St} = 0.150 \cdot (T - 10)$ and, when $T > 10\,^\circ$C, $f_{St} = -0.054 \cdot (T - 10)$.

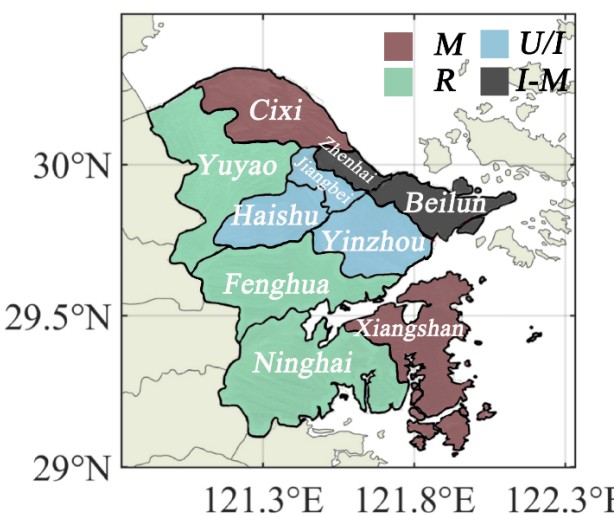

**Figure 7.** Atmospheric environment division of Ningbo's districts and counties.

The atmospheric environment data in the Ningbo area and the values of $D_0$ and $n$ are shown in Table 1. The data of the annual average temperature, annual average relative humidity and chloride ion deposition rate in each atmospheric environment were obtained from the China Meteorological Data System (http://data.cma.cn/, accessed on 1 July 2021) and National Materials Corrosion and Protection Data Center (https://www.corrdata. org.cn/, accessed on 1 July 2021). The sulfur dioxide concentration in Ningbo's districts and counties was specially commissioned at Qingdao Hengli Environmental Technology Research Institute Ltd. for testing (tested on 1 May 2021), and the testing locations were outdoor. As can be seen from the table, the sulfur dioxide concentration, chloride ion deposition rate, $D_0$ value and $n$ value are the maximum for the industrial–marine (I–M) atmosphere.

**Table 1.** Atmospheric data and $D_0$ and $n$ values in the Ningbo area.

| Atmospheric Environment | $T$ $^\circ$C | $RH$ % | $P_c$ µg/m$^3$ | $S_d$ mg/(m$^2 \cdot$d) | $D_0$ µm | $n$ |
|---|---|---|---|---|---|---|
| M | | | 7.0 | 127 | 57.50 | 0.49 |
| R | | | 5.0 | 50 | 35.70 | 0.37 |
| U/I | 17.4 | 72.5 | 49.0 | 60 | 62.35 | 0.39 |
| I-M | | | 49.0 | 127 | 79.14 | 0.49 |

The data in Table 1 was substituted into Equation (1), and the predicted curves of the long-term corrosion depth of carbon steel in each atmospheric environment in the Ningbo area were obtained (as shown in Figure 8). It can be seen the corrosion rate from the largest to the smallest: industrial–marine (I–M) atmosphere, marine (M) atmosphere, urban/industrial (U/I) atmosphere and rural (R) atmosphere. It is worth mentioning that the corrosion depth of carbon steel exceeds 1600 µm under the 100-year corrosion age of the I–M atmospheric environment, which indicates that, in such an atmospheric environment, transmission towers are the most vulnerable to corrosion and should be focused on research.

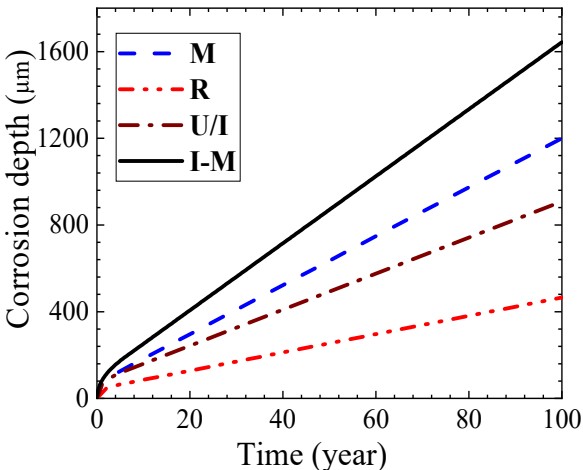

**Figure 8.** Long-term corrosion depth prediction curve for carbon steel in Ningbo.

## 3. Wind Climate Assessment of Transmission Tower Site

A wind climate assessment was carried out to determine the multidirectional extreme wind speed at the transmission tower site with different return periods. The Ningbo area belongs to the mixed wind climate region, which is influenced by East Asian monsoons and Northwest Pacific typhoons all year round. The daily maximum wind speed data (10-min time interval) and the corresponding wind direction records of the Ningbo ground meteorological station (58562 Yinzhou) from 1 January 1967 to 31 December 2019 were used as samples, and the data were extracted from the China Meteorological Data System (http://data.cma.cn/, accessed on 1 July 2021). According to the meteorological data specification, the collected wind speed data was carefully calibrated by adjusting the observation height, observation time interval and so on to the standard conditions. Figure 9 shows the division of 16 wind angles. D1 is the northerly wind direction, rotated clockwise every 22.5 ° as a wind direction, divided into D1, D2, . . . , D16. The observed daily maximum 10-min average wind speed series were categorized into 16 wind direction angles. By excluding the typhoon wind speed data, the annual maximum wind speed series of constant winds in each wind direction can be filtered. Figure 10 shows the annual maximum wind speed series of constant winds in wind direction D14.

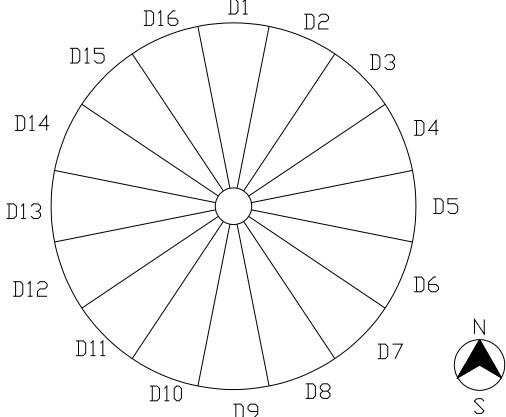

**Figure 9.** Definition of the wind direction azimuth.

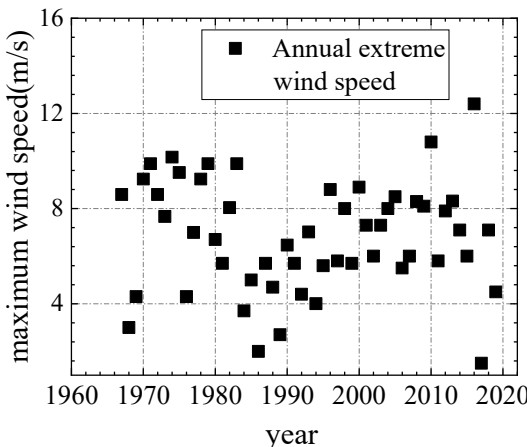

**Figure 10.** Annual maximum wind speed series of the constant winds in wind direction D14.

Due to the limitation of the sample size of historical typhoon wind speed observations, this paper adopted the typhoon full-track simulation and applied the Yan Meng wind field model to obtain the 600-year near-surface typhoon wind speeds affecting the Ningbo area [31]. The full-track simulation process of a single typhoon is shown in Figure 11, which can be divided into five steps, i.e., establishing the genesis, track and intensity models, calibrating the track and intensity simulation results and activating the typhoon wind field model. The Vickery empirical model [32] was used to determine the values of the key wind field parameters, such as the maximum wind speed radius $R_{max}$ and Holland pressure profile parameter $B$. The roughness length was taken as 0.05 m. The 600-year typhoon wind speed series obtained from the simulation was filtered and categorized according to the definition of the wind azimuth in Figure 9, and the annual maximum typhoon wind speed series under wind direction D14 is shown in Figure 12.

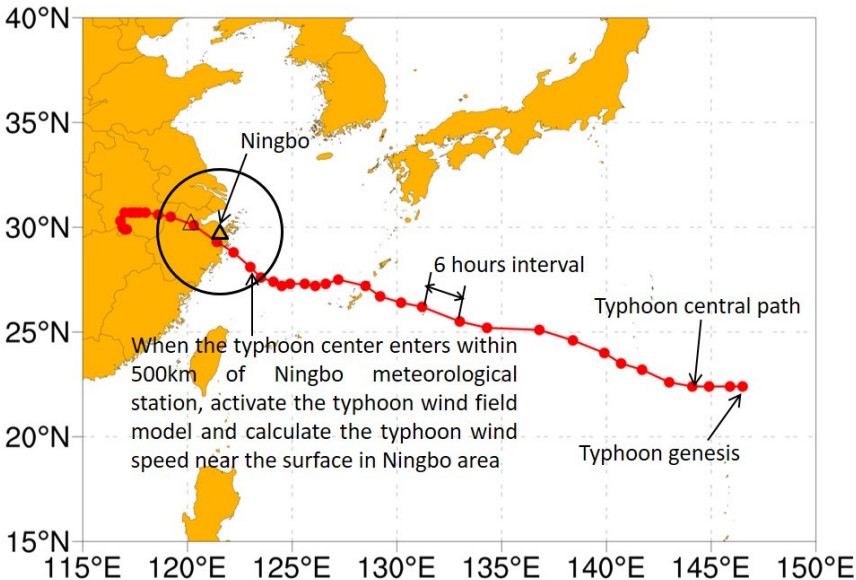

**Figure 11.** Flow chart of the full-track simulation of a single typhoon.

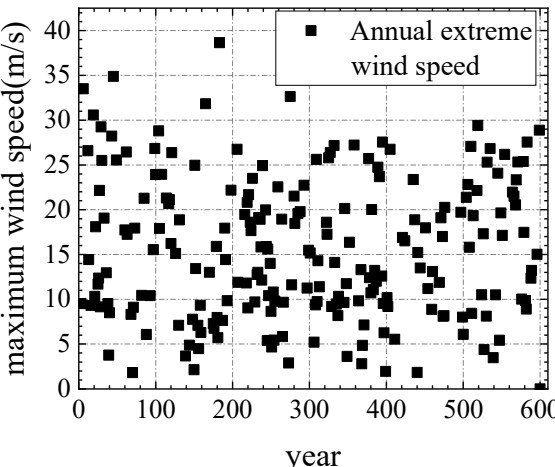

**Figure 12.** Annual maximum wind speed series of a typhoon in wind direction D14.

Based on the *t*-Copula function, the joint probability distribution models of the annual maximum wind speed series of constant wind and typhoons under 16 wind directions were constructed, respectively, and the multidirectional extreme wind speeds under different return periods were calculated by using this joint distribution model, and the specific procedures can be referred to the literature [33]. Figure 13 shows the marginal probability distribution of the annual maximum wind speed in wind direction D14. It can be seen that this wind direction is dominated by typhoons. In fact, all 16 wind directions in the Ningbo area are controlled by typhoons. In this paper, the parameters of the *t*-Copula function were estimated by the Maximum Likelihood Estimation (MLE). Table 2 shows the correlation coefficients of the adjacent wind directions in the correlation matrix. It can be seen that the correlation coefficients of the adjacent wind directions of the typhoons in the Ningbo area all exceed 0.7. The correlation coefficient of the adjacent wind directions between wind direction D2 and wind direction D8 even reaches 0.99, which proves that the correlation of the adjacent wind directions is very significant. The joint probability density contours of the typhoons under wind directions D13 and D14 are shown in Figure 14. It can be found that the extreme typhoon wind speeds under wind directions D13 and D14 are positively correlated under the consideration of wind direction correlation, while they do not show the correlation under the disregard of the wind direction correlation.

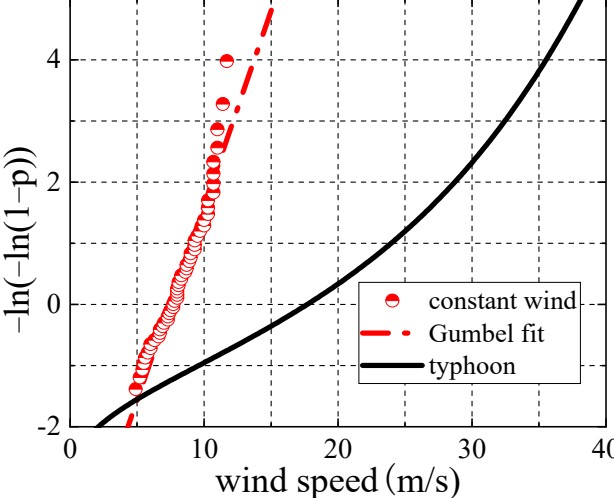

**Figure 13.** The marginal probability distribution of the annual maximum wind speed in wind direction D14.

**Table 2.** Correlation coefficients of the adjacent wind directions of typhoons.

| wind direction | 1–2 | 2–3 | 3–4 | 4–5 | 5–6 | 6–7 | 7–8 | 8–9 |
|---|---|---|---|---|---|---|---|---|
| correlation coefficients | 0.75 | 0.99 | 0.99 | 0.99 | 0.99 | 0.99 | 0.99 | 0.85 |
| wind direction | 9–10 | 10–11 | 11–12 | 12–13 | 13–14 | 14–15 | 15–16 | 16–1 |
| correlation coefficients | 0.89 | 0.86 | 0.82 | 0.85 | 0.88 | 0.89 | 0.92 | 0.86 |

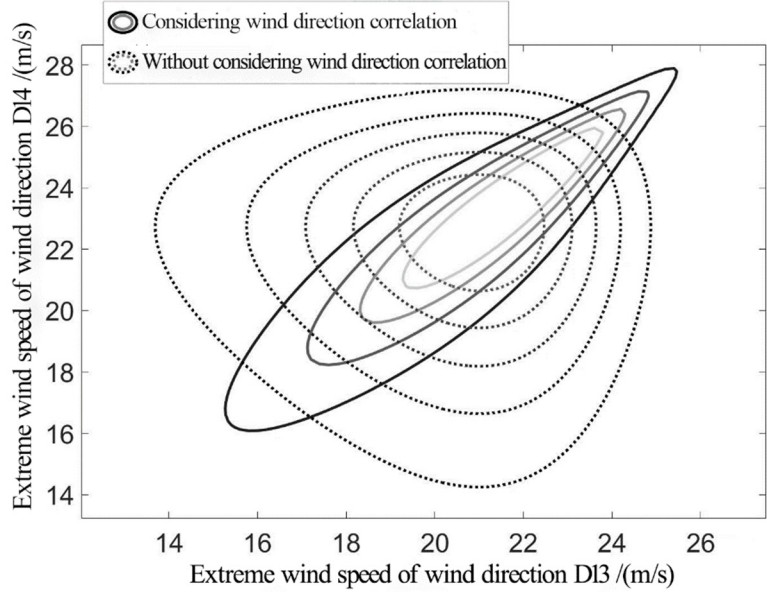

**Figure 14.** The joint probability density contours of the typhoons under wind directions D13 and D14.

Figure 15 shows the comparison between the 10/50-year extreme wind speeds and the design wind speed given by the code in the Ningbo area. It can be found that the extreme wind speed of a typhoon is significantly greater than the extreme wind speed of a constant wind in each wind direction, indicating that the wind climate in the Ningbo area is mainly controlled by typhoons. The design wind speed given by the code is on the risky side. In addition, by comparing the extreme wind speed of each wind direction azimuth, it can be found that wind direction D2 is the most unfavorable wind direction, and its 50-year extreme wind speed value is 39.5 m/s.

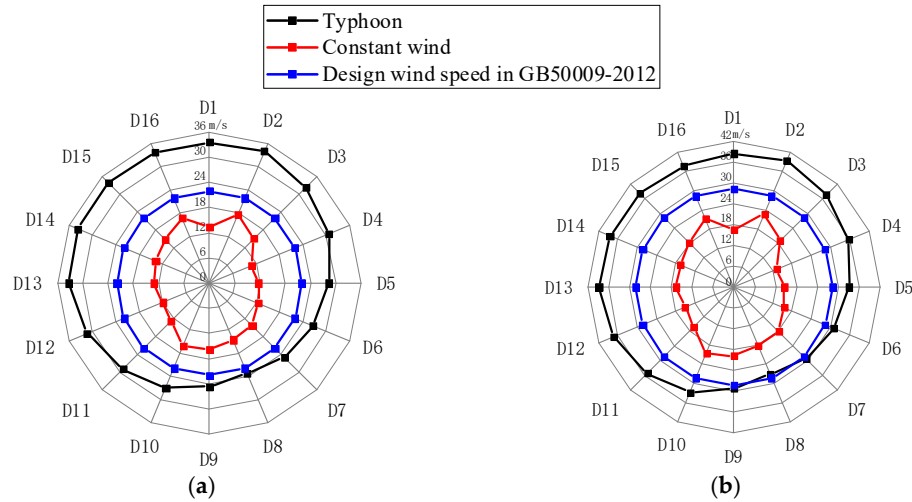

**Figure 15.** Comparison of the extreme wind speed with the design wind speed given by the code in Ningbo. (**a**) Ten-year return period. (**b**) Fifty-year return period.

## 4. Wind-Induced Fragility Analysis of Corroded Transmission Tower

### 4.1. Finite Element Model of Corroded Transmission Tower

A typical 110 KV cathead linear tower in Ningbo Zhenhai (industrial–marine atmospheric environment) was taken as the research object, with a tower height of 35.5 m, a nominal height of 30 m and a bottom root opening of 6.509 m, and its single line diagram is shown in Figure 16a. Using ANSYS to establish its finite element model, the tower main material, slant material, cross-partition material and auxiliary material were used to establish the BEAM188 unit. Among them, the main material was Q345 steel, and the rest of the material was Q235 steel; the steel density was 7850 kg/m$^3$, the Poisson's ratio was 0.3 and the elastic modulus was 206 GPa. Notably, Q345 is a kind of low-alloy steel, as defined in the Chinese steel specification, where "Q" means the yield strength, and "345" means the yield strength of this steel is 345 MPa. Q235 is a common carbon structural steel, and "235" represents the yield value of this steel, which is around 235 MPa. The finite element model is shown in Figure 16b. The first two fundamental mode shapes of the transmission tower without rust are depicted in Figure 16c,d, where it can be seen that the first two modes are dominated by the sway component.

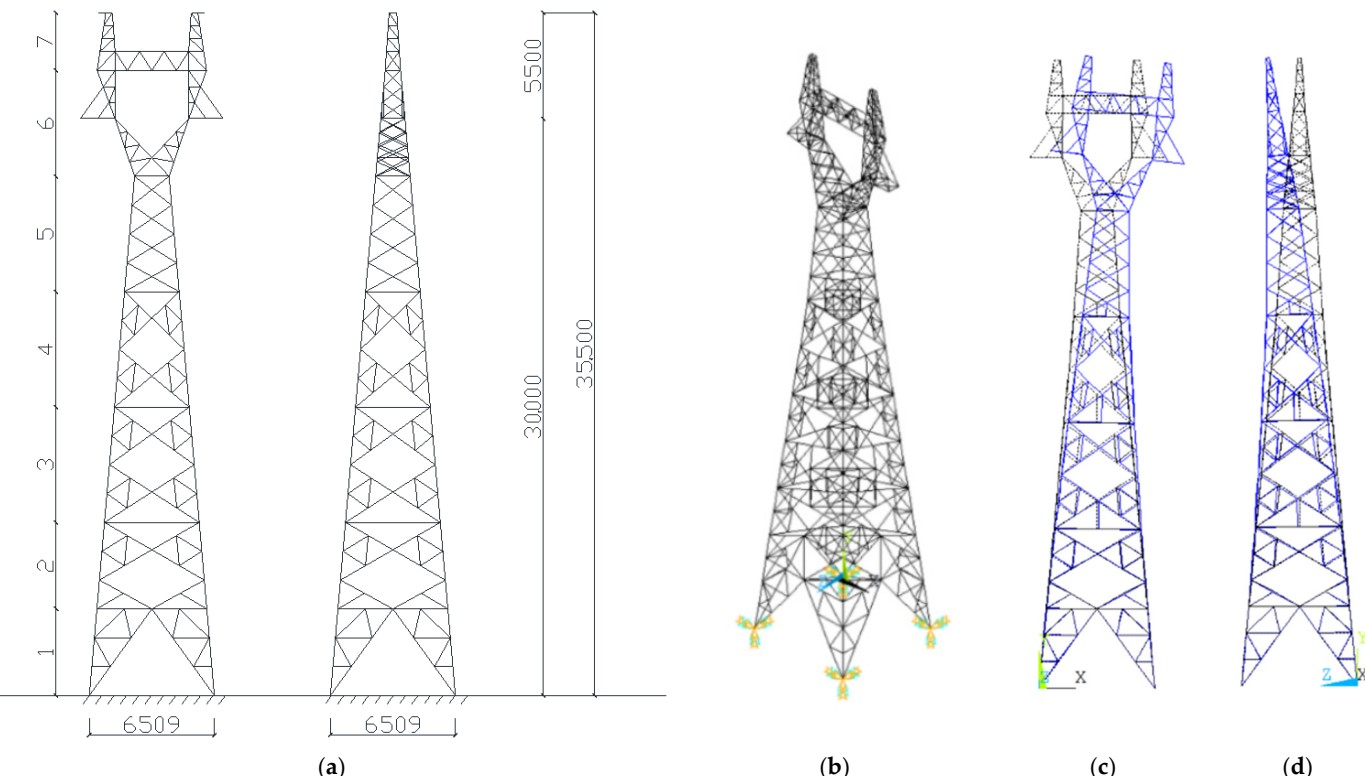

**Figure 16.** Schematic diagram of the transmission tower. (**a**) Single line diagram (unit: mm); (**b**) finite element model; (**c**) 1st mode f$_1$ = 3.33 Hz; (**d**) 2nd mode f$_2$ = 3.40 Hz.

In actual engineering, steel transmission towers will receive anticorrosive treatment during the early stage of construction, and the galvanized layer is usually painted on the surface. However, with the increase of the service life, the galvanized layer gradually loses its protective ability due to external erosion. Therefore, this paper took the time when the transmission tower loses the protection of the galvanized layer as the initial year and established models of the steel transmission tower under 30/60/90 corrosion years, respectively. The effect of steel corrosion on the deterioration of the wind resistance of the transmission tower was simulated by weakening the cross-sectional dimensions of the members. It should be noted that only the uniform corrosion of the whole tower was considered in this paper.

Figure 17 shows the cross-section of the corroded angle. *D* represents the initial section thickness; *D'* represents the residual depth of corrosion. Based on the prediction results of Figure 8, the corrosion depth of carbon steel under the I–M atmospheric environment for 30, 60 and 90 years of corrosion can be calculated. By subtracting the corrosion depth value from the initial section thickness *D* of each angle member, the residual corrosion depth *D'* of the angle sections can be obtained, as shown in Table 3. By integrating *D'* into ANSYS, finite element models of the steel transmission tower under 30/60/90 corrosion years were finally established.

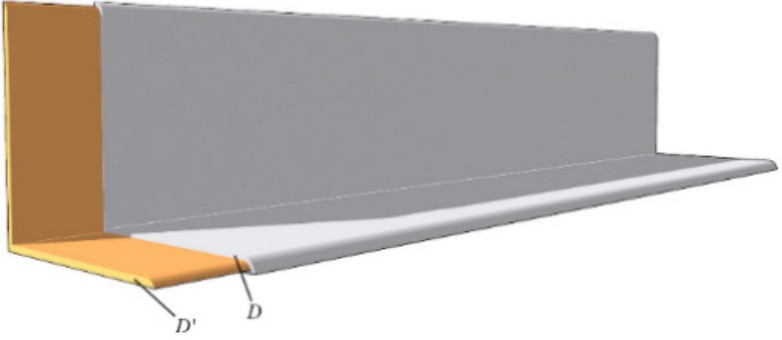

**Figure 17.** Cross-section of a corroded angle.

**Table 3.** Residual depth of corrosion *D'* of the angle sections in the I–M atmosphere (unit: mm).

| Steel Type | Angle Size | 30 Years | 60 Years | 90 Years |
|---|---|---|---|---|
| Q345 | L90 × 8 | 7.44 | 6.98 | 6.51 |
|  | L100 × 8 | 7.44 | 6.98 | 6.51 |
| Q235 | L40 × 4 | 3.44 | 2.98 | 2.51 |
|  | L45 × 4 | 3.44 | 2.98 | 2.51 |
|  | L50 × 5 | 4.44 | 3.98 | 3.51 |
|  | L56 × 5 | 4.44 | 3.98 | 3.51 |
|  | L63 × 5 | 4.44 | 3.98 | 3.51 |
|  | L70 × 6 | 5.44 | 4.98 | 4.51 |

### 4.2. Construction of Fragility Model

The wind-induced fragility function reflects the probability that the structure reaches or exceeds a certain limit state under extreme wind loads, and the failure probability when the wind load effects $S_D$ exceed the structural bearing capacity $R_C$ can be determined by the following formula:

$$P_f = P_r(\frac{R_C}{S_D} \leq 1) \tag{4}$$

In the above equation, assuming that both $S_D$ and $R_C$ obey lognormal distribution, and selecting the maximum displacement at the top of the tower as the index of the load effects, the above equation can be transformed into:

$$P_f = \Phi\left[\frac{\ln(m_D) - \ln(m_C)}{\sqrt{\beta_C^2 + \beta_D^2}}\right] \tag{5}$$

where: $\Phi[\bullet]$ denotes the standard normal distribution function, $m_D$ is the maximum displacement at the top of the transmission tower under the given wind loads, $m_C$ is the structural bearing capacity under different limit states, $\beta_C$ denotes the logarithmic standard deviation of the structural bearing capacity and the value can be taken from the literature [10]. $\beta_D$ denotes the logarithmic standard deviation of the wind load effects.

### 4.3. Wind Load Simulation and Analysis of Wind Load Effects

The transmission tower was divided into six simulated segments from bottom to top, and the wind blocking area of each segment was calculated separately. The wind speed at the vertical midpoint of each segment was used as the simulated wind speed of the corresponding segment, and the simulated segments were numbered in increasing order from bottom to top, i.e., segment 1 to segment 6, as shown in Figure 18. According to the wind climate assessment results, the extreme wind speed under the most unfavorable wind direction (D2) in the Ningbo area was selected as the basic wind speed, the ground roughness category was set to class B, the wind profile index was set to 0.16, the Kaimal spectrum was used to simulate the wind speed spectrum, the harmonic superposition method was used to simulate and generate the wind speed series in the downwind direction and the wind load series was generated according to the technical code: DL/T5154-2012 [34].

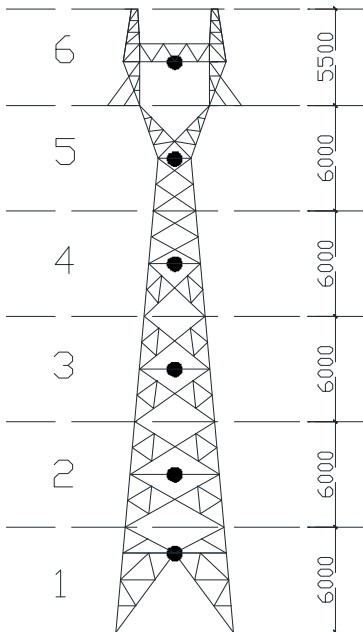

**Figure 18.** Schematic diagram of the simulated wind loads in sections (unit: mm).

In the wind load effect analysis of this paper, the incremental dynamic analysis method was used to set the basic wind speed from 30 m/s and increase 5 m/s each time, setting nine groups of basic wind speeds (i.e., 30 m/s, 35 m/s, 40 m/s, 45 m/s, 50 m/s, 55 m/s, 60 m/s, 65 m/s and 70 m/s). Considering the randomness of the wind loads, 10 sets of wind load series were generated at the same basic wind speed, and 10 sets of structural dynamic responses were estimated by performing linear elastic analyses in the time domain in ANSYS, respectively, and then, the analytical functions of the wind speed and the index of the load effects, i.e., maximum displacement at the top of the tower, were established by the regression analysis.

The relationship between $m_D$ and the basic wind speed $v$ generally obeys the power exponential relationship [10], and its logarithmic expression is:

$$\ln m_D = a + b \ln(v) \tag{6}$$

where the values of the coefficients $a$ and $b$ can be determined by a regression analysis. Figure 19 shows the wind load effect functions of the transmission tower under different corrosion ages. The corresponding analytical functions are also presented in the figure. It can be seen that the relationship between the maximum displacement at the top of the tower and the basic wind speed conform to the power exponential relationship.

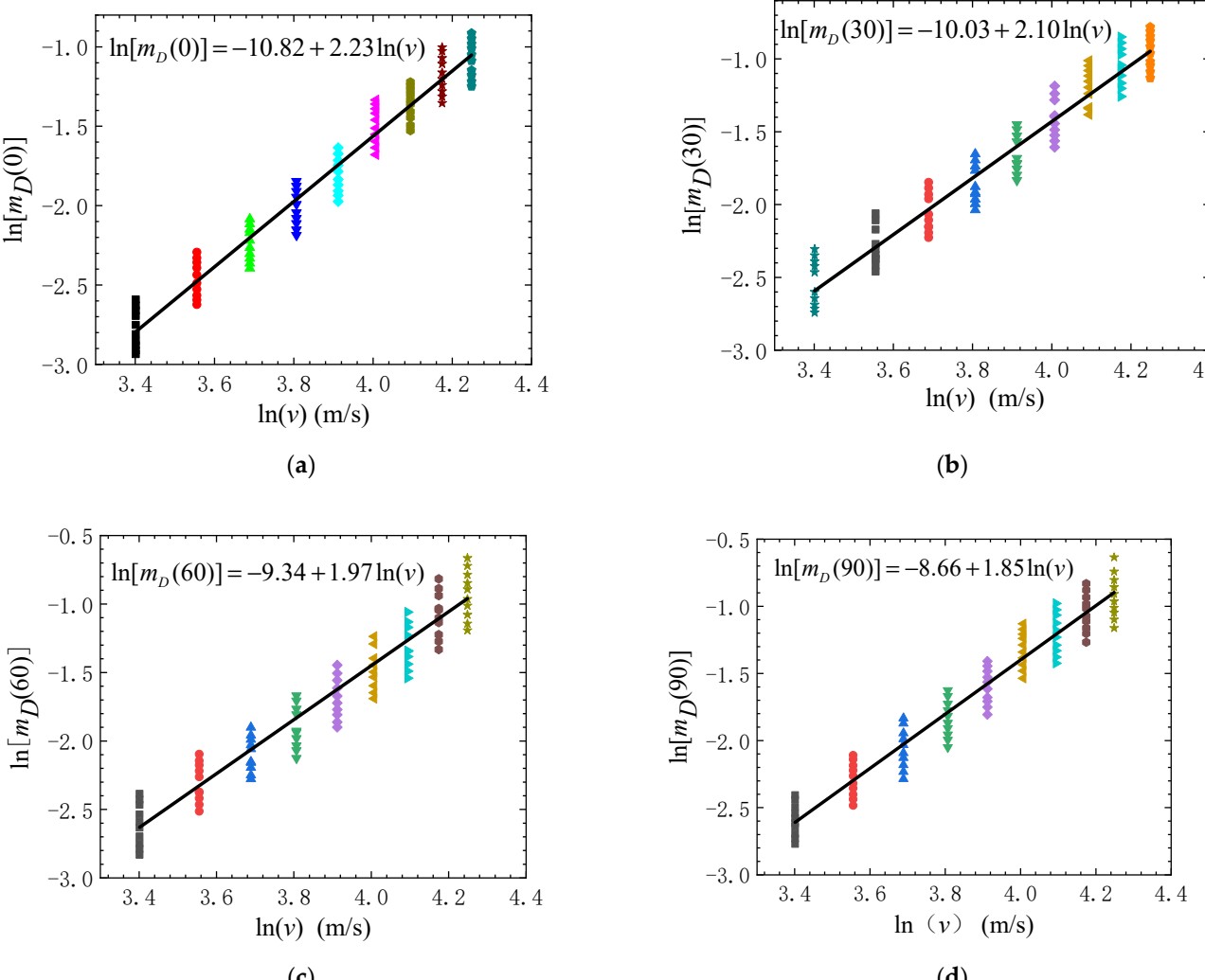

**Figure 19.** Wind load effect functions of the transmission tower under different corrosion ages:
(**a**) 0 year; (**b**) 30 years; (**c**) 60 years; (**d**) 90 years.

### 4.4. Definition of Structural Limit Damage Modes

Pushover analyses were performed on the finite element models of the transmission tower with different corrosion ages, and the inverse triangular distributed lateral loading mode was used to obtain pushover curves between the base shear and the top maximum displacement, as shown in Figure 20. The maximum displacements at the top in the elastic stage for the transmission tower with corrosion ages of 0, 30, 60 and 90 years are all at 0.4 m, which is much larger than the displacement limit given by the code [34] (the code limit is 3H/1000, i.e., 0.1 m). From Figure 20, it can also be seen that the ultimate displacements of the transmission tower with different corrosion ages are all at 1.86 m. Based on the above analysis, and according to the research results related to the transmission tower damage modes and failure laws [9–11], three types of damage modes were defined in this paper, i.e., minor damage, moderate damage, severe damage and collapse, and the quantitative index limits for each damage mode of the corroded transmission tower were determined, as shown in Table 4.

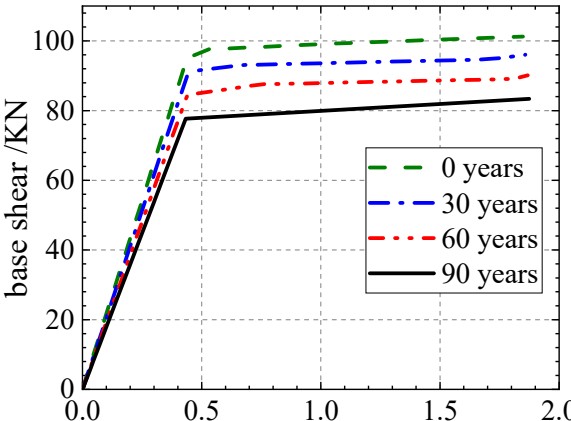

**Figure 20.** Pushover curves of the transmission tower under different corrosion ages.

**Table 4.** Index limits for each damage mode of the transmission tower.

| Performance Level | Basically Intact | Minor Damage | Moderate Damage | Severe Damage and Collapse |
|---|---|---|---|---|
| Maximum displacement of tower top (m) | $m_C < 0.20$ | $0.20 \leq m_C < 0.4$ | $0.4 \leq m_C < 1.86$ | $m_C \geq 1.86$ |

*4.5. Fragility Evaluation*

Substituting the wind load effect functions and the index limits into Equation (5), the wind-induced fragility curves of the transmission tower under an industrial–marine atmospheric environment with corrosion ages of 0, 30, 60 and 90 years can be obtained, respectively, as shown in Figure 21. It can be seen that the nominal mechanical properties of steel decrease as the corrosion age increases, leading to a significant increase in the probability of minor damage, moderate damage and severe damage of the transmission tower, and the deterioration of the wind resistance of the transmission tower by corrosion will be amplified under the effect of extreme wind damage. Under the wind loads of 50-year return periods in the most unfavorable wind direction (the corresponding typhoon wind speed is 39.5 m/s) in the Ningbo area, the tower will hardly experience severe damage and collapse, the probability of moderate damage is within 10%, and the probability of minor damage is controlled between 10% and 40%.

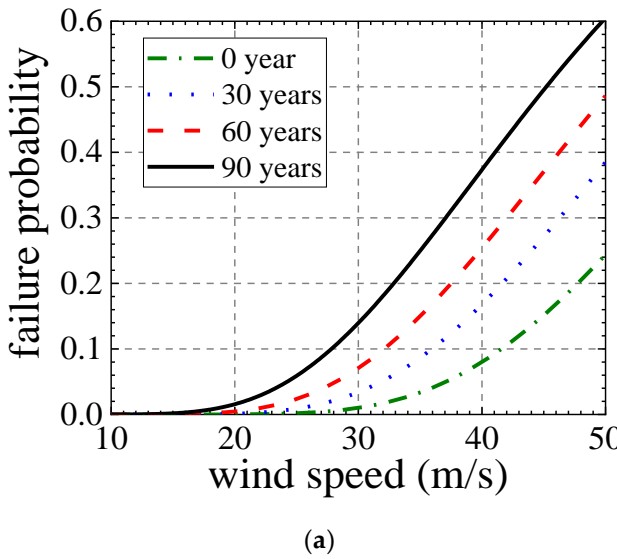

(a)

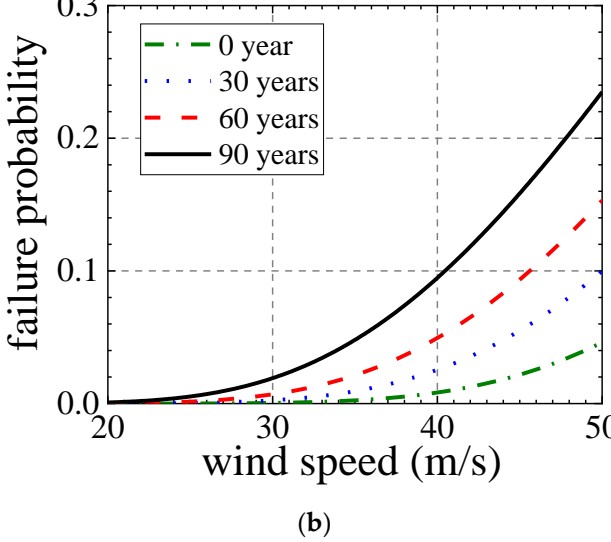

(b)

**Figure 21.** *Cont.*

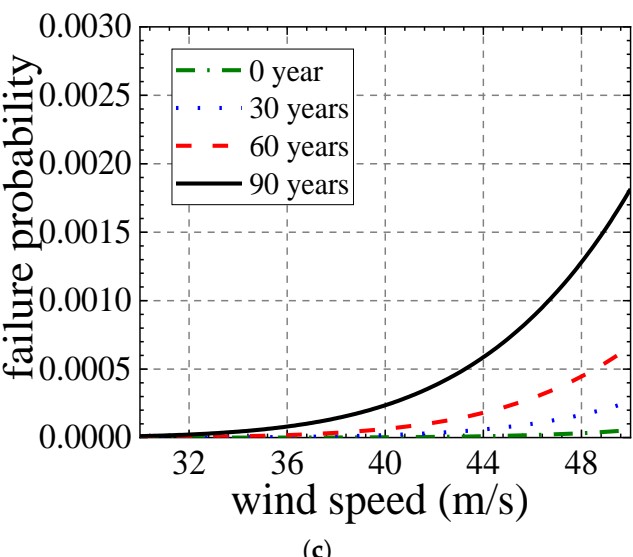

(**c**)

**Figure 21.** Wind-induced fragility curves of the transmission tower with different corrosion ages under the industrial-marine atmospheric environment in the Ningbo area. (**a**) Minor damage. (**b**) moderate damage and (**c**) severe damage and collapse.

## 5. Conclusions

In this paper, a long-term corrosion depth prediction model for carbon steel applicable of the Ningbo area was established, a mixed climate multi-wind direction extreme wind speed estimation method was developed and wind-induced fragility curves for the transmission tower under different corrosion ages were obtained, which can be used to quantitatively evaluate the effect of long-term corrosion on the deterioration of wind resistance of a steel transmission tower during its service life. The specific findings are as follows:

(1) The corrosion rates of carbon steel in different atmospheric environments in the Ningbo area are in descending order: industrial–marine (I–M) atmosphere, marine (M) atmosphere, urban/industrial (U/I) atmosphere and rural (R) atmosphere. Among them, the corrosion depth of carbon steel exceeds 1600 μm under the 100-year corrosion age of the I–M atmospheric environment, which indicates that, in such an atmospheric environment, transmission towers are the most vulnerable to corrosion;

(2) The wind climate in the Ningbo area is mainly controlled by typhoons, and the design wind speed given by the current load code is on the risky side;

(3) With the increase of the corrosion ages, the nominal mechanical properties of steel decrease, making the probability of minor damage, moderate damage and severe damage to the transmission tower increase. Under the wind loads of 50-year return periods in the most unfavorable wind direction in the Ningbo area, the tower will hardly experience severe damage and collapse, the probability of moderate damage is within 10% and the probability of minor damage is controlled between 10% and 40%.

It is worth mentioning that the maximum displacement of the top of the transmission tower was selected as the load effect index in this paper, and the probability of the maximum displacement of the top exceeding the limit value under different corrosion years was taken as the failure probability. In fact, there are various structural failure modes of transmission towers under wind loads, and wind-induced fatigue is one of the important failure factors. It is of great practical significance to conduct a structural wind-induced fatigue analysis based on the cumulative damage theory and probabilistic fracture mechanics theory for the transmission tower structures in future studies.

**Author Contributions:** Conceptualization, Q.L.; Writing—original draft preparation, H.J. and Q.Q.; Writing—review and editing, Q.L. and Y.L.; Supervision, J.Z., J.M., W.F. and M.H. All authors have read and agreed to the published version of the manuscript.

**Funding:** This research was funded by The National Natural Science Foundation of China (51908496 and 51820105012), The Natural Science Foundation of Zhejiang Province and Ningbo City (LQ20E080001 and 2021J168) and the Science and Technology Special Project of Ningbo Fenghua District (202008502).

**Institutional Review Board Statement:** Not applicable.

**Informed Consent Statement:** Not applicable.

**Data Availability Statement:** Some or all the data and models that support the findings of this study are available from the corresponding author upon reasonable request.

**Acknowledgments:** We thank the Ningbo Kaihong Engineering Consulting Co. and State Grid Ningbo Power Supply Company for providing the research objects for this paper.

**Conflicts of Interest:** The authors declare no conflict of interest.

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
