# Peer review of "Typhoon-Induced Fragility Analysis of Transmission Tower in Ningbo Area Considering the Effect of Long-Term Corrosion"

_applsci, doi:10.3390/app12094774_

Round 1

Reviewer 1 Report

In the present study, authors present good results on typhoon-induced fragility analysis of transmission tower. Based on finite element method, corrosion years of transmission tower were predicted. It is helpful to understand/predict that effects on the environment threatening the life year of transmission tower. Before the paper being accepted, please revise an issue. Please use symbol “μm”, not “um” in manuscript, including Figures.

Reviewer 2 Report

This manuscript sought to build an analytical model for long-term corrosion depth prediction of carbon steel based on the local atmospheric environment data, using the typhoon full-track simulation and wind field simulation. The manuscript is very well written and presented. I suggest accepting the manuscript in its present form.

  1. a small typo in line 55. pH is written PH.

Reviewer 3 Report

This manuscript presents a long term corrosion depth prediction model for carbon steel and a typhoon induced fragility analysis of transmission tower, enabling the evaluation of the effect of long term corrosion on the deterioration of wind resistance of a steel transmission tower during its service life. After the minor revision comments below are addressed, the paper is recommended for publishing.

  1. Line 132, for long duration wind events such as typhoons, cumulative damage mechanisms, e.g., ratcheting, and low cycle fatigue, may occur. The static pushover analysis, however, is not capable of capturing these failure mechanisms. Please discuss this aspect in the manuscript.
  2. Line 164, please clarify what D is in the equation? Is it D1, D2, the difference between D1 and D2 or the total corrosion depth?
  3. Line 321&322, please define what Q345 and Q235 steel are, including material properties like the yield stress.
  4. Section 4.3, line 374-378, please clarify how structural dynamic responses were estimated. Is linear elastic analysis or nonlinear analysis adopted?
  5. Section 4.3, Figure 19, how were the 9 different wind speeds selected?
  6. Section 4, what were the uncertainties considered in the fragility analysis? Please discuss.

Reviewer 4 Report

This is a very important article to relate seemingly irrelevant causes for the corrosion effects of the transmission tower. It is true that in a complex system when in doubt, a power-law distribution can be utilized to relate two factors that cannot be directly established via first principles. It is true for Paris Law for fatigue crack propagation and Basquin's Law for finite fatigue life predictions.

However, before the archival publication of this article, an in-depth study of material failure modes must be presented and prioritized. For example, due to wind-structure interactions, what is the typical frequency under the Typhoon versus the normal wind speed below 20mph.  How is the fatigue in comparison with the corrosion effect? Do we need to use Goodman or Soderberg line for infinite life span prediction? 

Round 2

Reviewer 4 Report

The paper has a lot of significant data pertaining to corrosion and the related life span and base stress predictions. The paper is ok for publication.